

# A survey of the views and capabilities of community pharmacists in Western Australia regarding the rescheduling of selected oral antibiotics in a framework of pharmacist prescribing

Fatima Sinkala[1], Richard Parsons[1], Bruce Sunderland[1], Kreshnik Hoti[2] and Petra Czarniak[1]

[1] School of Pharmacy, Curtin University, Perth, Western Australia, Australia
[2] Faculty of Medicine, Division of Pharmacy, University of Prishtina, Pristina, Kosovo

Corresponding author
Petra Czarniak,
P.Czarniak@curtin.edu.au

## ABSTRACT

**Background.** Antibiotic misuse in the community contributes to antimicrobial resistance. One way to address this may be by better utilizing community pharmacists' skills in antibiotic prescribing. The aims of this study were to examine the level of support for "down-scheduling" selected antibiotics and to evaluate factors determining the appropriateness of community pharmacist prescribing for a limited range of infections, including their decision to refer to a doctor.

**Methods.** Self-administered questionnaires, including graded case vignette scenarios simulating real practice, were sent to Western Australian community pharmacists. In addition to descriptive statistics and chi-square testing, a General Estimating Equation (GEE) was used to identify factors associated with appropriateness of therapy and the decision to refer, for each of the seven vignettes.

**Results.** Of the 240 pharmacists surveyed, 90 (37.5%) responded, yielding 630 responses to seven different case vignettes. There was more than 60% respondent support for expanded prescribing (rescheduling) of commonly prescribed antibiotics. Overall 426/630 (67.6%) chose to treat the patient while the remaining 204/630 (32.4%) referred the patient to a doctor. Of those electing to treat, 380/426 (89.2%) opted to use oral antibiotics, with 293/380 (77.2%) treating with an appropriate selection and regimen. The GEE model indicated that pharmacists were more likely to prescribe inappropriately for conditions such as otitis media ($p = 0.0060$) and urinary tract infection in pregnancy ($p < 0.0001$) compared to more complex conditions. Over 80% of all pharmacists would refer the patient to a doctor following no improvement within 3 days, or within 24 h in the case of community acquired pneumonia. It was more common for younger pharmacists to refer the patient to a doctor ($p = 0.0165$).

**Discussion.** This study adds further insight into community pharmacy/pharmacist characteristics associated with appropriateness of oral antibiotic selection and the decision to refer to doctors. These findings require consideration in designing pharmacist over-the-counter prescribing models for oral antibiotics.

## INTRODUCTION

Over the last 25 to 30 years, a number of countries including New Zealand (NZ), the United States of America (USA), the United Kingdom (UK), the Netherlands, Japan and Australia, have reclassified or switched several drugs with an established safety profile from prescription to non-prescription availability (*Gauld et al., 2014*; *Australian Self-Medication Industry, 2017*). In Australia, examples of drugs that have been reclassified include emergency hormonal contraception, the oral azole antifungal fluconazole and proton pump inhibitors (*Gauld et al., 2014*). In NZ and the UK, the antibiotics trimethoprim and azithromycin, respectively, may also be supplied by suitably trained pharmacists (*Aronson, 2009*; *Gauld et al., 2017*). As antibiotic resistance has been declared a global threat by the World Health Organization (WHO) (*Huttner et al., 2013*), employing pharmacists' knowledge and expertise in the appropriate selection and use of reclassified antibiotics could potentially help reduce the level of inappropriate antibiotic use and therefore resistance (*Booth et al., 2013*). Antimicrobial agents available under strict protocols from suitably qualified pharmacists to maintain antimicrobial stewardship could improve patient access to immediate treatment and reduce the workload of general practitioners (GPs) (*Gauld et al., 2017*; *Booth et al., 2013*; *Reeves et al., 1999*). Widespread inappropriate use of antibiotics in hospitals and the community has led to the development of a specialist antibiotic pharmacist's role in the UK (*Weller & Jamieson, 2004*). In a study in the community pharmacy setting, researchers in the USA investigated the use of rapid point-of-care tests by pharmacists to allow clinical decision making so that appropriate treatment could be initiated for patients with influenza or group A *Streptococcus* (GAS) pharyngitis. Researchers reported that this innovative physician-pharmacist disease management program produced positive progress toward reducing the inappropriate use of antibiotics (*Klepser, Adams & Klepser, 2015*).

Pharmacists are reported to adhere to prescribing guidelines (*Tonna et al., 2010*) and improve patients' access to medicines (*Hale et al., 2016*). In Australia, the current framework administering access to medicines includes "down-scheduling" and selected medicines previously restricted to 'Prescription only' or 'Schedule 4 (S4)' have been reclassified as 'Pharmacist only' or 'Schedule 3' (S3) medicines, thereby allowing them to be provided over-the-counter (OTC) (*Australian Self-Medication Industry, 2017*). It is a requirement in Australia that S3 medicines are provided by a pharmacist or under the direct supervision of a pharmacist (*Samsom, 2018*). In 2010, chloramphenicol ophthalmic products were reclassified from S4 to S3 thereby expanding the options and capability of Australian pharmacists treating acute bacterial conjunctivitis and providing improved community access to an effective antibiotic treatment. Similar rescheduling decisions had been made in the UK and NZ in previous years (*Alkhatib et al., 2015*).

The roles of community and hospital pharmacists have been extended to include prescribing, in several countries including Canada (*Law et al., 2012*; *Lynas, 2007*), the USA (*Zellmer, 1995*), NZ and the UK, where pharmacists are working within various prescribing models, including collaborative, supplementary and independent pharmacist prescribing (*Tonna et al., 2007*). In Australia, expanding the pharmacist's prescribing

role is still under review (*Hughes et al., 2014*; *Hoti et al., 2010*; *Kay, Bajorek & Brien, 2006*; *Hoti, Hughes & Sunderland, 2014*; *Hoti, Hughes & Sunderland, 2011*). Currently, a range of Schedule 2 (S2) drugs which are only available from pharmacies (Pharmacy only) and S3 drugs which require the direct involvement of a pharmacist (Pharmacist only), are available from community pharmacies for minor or self-limiting conditions (*Hoti et al., 2010*; *Paudyal et al., 2013*; *Paudyal et al., 2012*). A number of protocols are in place on selected 'Pharmacist only' medicines such as prescribing emergency contraception (*Samsom, 2018*). Access to selected antibiotics via a protocol could enable community pharmacists to effectively treat a range of infections (*Hoti et al., 2010*; *Paudyal et al., 2013*; *Paudyal et al., 2012*).

There are additional factors in the health system in Australia that influence patient access to antibiotics. A repeat prescribing model is part of the Pharmaceutical Benefits Scheme in Australia, whereby a medical practitioner can order repeats for antibiotics initially prescribed by them. The repeat antibiotic prescription is intended to be obtained in case the patient/client does not adequately improve following the initial course of antibiotics, or if a relapse of the same symptoms subsequently occurs. These repeat prescriptions are valid within a period of 12 months following the initial prescription (*Zayegh et al., 2014*). This may lead to misuse of antibiotics in patients/clients with repeat prescriptions that choose to self-diagnose and self-manage their symptoms. To address this issue, expanding pharmacists' role in antibiotic prescribing has been suggested (*Zayegh et al., 2014*).

Given that much of the existing literature has explored pharmacists' views on expanding their prescribing role (*Tonna et al., 2007*; *Hoti et al., 2010*; *Hoti, Hughes & Sunderland, 2014*) and support has been given for pharmacist prescribing for a limited range of infections, there is a need for research aimed at assessing pharmacists' perceptions when confronted with various real life scenarios of antibiotic prescribing (*Tonna et al., 2007*; *Hoti et al., 2010*; *Kay, Bajorek & Brien, 2006*; *Hoti, Hughes & Sunderland, 2014*; *Res, Hoti & Charrois, 2017*). This would provide valuable insight to policymakers in relation to designing a model of pharmacist prescribing of antibiotics for a limited range of infections in the community setting. This study therefore aims to explore factors determining the appropriateness of community pharmacist prescribing for a limited range of infections, including their decision to refer to a doctor and examine the level of support for "down-scheduling" selected antibiotics.

## MATERIALS AND METHODS

This cross-sectional quantitative study involved a postal survey of practising rural and metropolitan community pharmacists in Western Australia (WA). Questions included seven case vignettes which were used due to their ability to simulate key features of a range of 'real-life' scenarios (*Evans et al., 2015*; *Atzmüller & Steiner, 2010*). They were chosen because they carried the external validity strengths of quantitative-survey based research as well as the internal validity strengths of experimental methods (*Evans et al., 2015*; *Atzmüller & Steiner, 2010*). A review of the literature informed the design of the questionnaire (*Alkhatib et al., 2015*; *Evans et al., 2015*; *Atzmüller & Steiner, 2010*), while the medical conditions shown by scenarios in the vignettes were based on literature and
by using the Australian Therapeutic Guidelines (ATG) for antibiotics (*Evans et al., 2015*; *Antibiotic Expert Groups, 2014*). The questionnaire and vignettes, which were developed by the researchers, were piloted by six community pharmacists, some with extensive antibiotic experience, for face and content validity and their feedback was used to improve the questionnaire and vignettes.

The sections of the questionnaire were: (A) demographic information; (B) statements of views on expanding the pharmacist's role in prescribing antibiotics and (C) included the case vignettes.

## Case vignettes

The case vignettes consisted of seven scenarios and the respondents were asked for their preferred treatment option, under the hypothetical assumption that they were permitted to prescribe oral antibiotics. The design of the case vignettes was such that the final diagnosis was evident. The scenarios included consideration to some of the key features of case-vignette design i.e., experimental aspect (various antibiotic based scenarios and their effect on respondent's choice of treatment); controlled aspect (i.e., same pharmacists responding to different scenarios) and contextual aspect as demonstrated by variability within each of the scenarios allowing for the verisimilitude of the scenario (*Evans et al., 2015*). The respondents were asked to select an option from: refer to a GP, treat with oral antibiotics (from a list), or choose a different treatment regimen. The list included antibiotic regimens recommended in the ATG Antibiotics (*Evans et al., 2015*; *Antibiotic Expert Groups, 2014*). The vignettes were graded according to disease complexity. In cases where the selected management option was not to refer to a GP, the respondent was asked what action they would take if there was no improvement following 24 h (community acquired pneumonia (CAP)) or three days (other vignettes).

## Questionnaire distribution

The sampling frame was the list of 434 metropolitan and 164 regional community pharmacies available from The Pharmacy Registration Authority of WA. A 40% sample was randomly selected using a web based randomizer. Hospital pharmacies in WA were excluded. The final questionnaire was distributed to 66 regional and 174 metropolitan community pharmacies in WA in March 2015 and coded to be able to identify non-respondents. A cover letter explaining the objectives and importance of the study was addressed to the 'manager/proprietor' and enclosed with the questionnaires, information sheet and a reply paid envelope. Reminder letters accompanied by additional questionnaires and reply paid envelopes were sent to non-responders in April 2015.

## Sample size determination

A sample size of 96 was considered the minimum necessary to conduct the inferential statistics to identify independent variables exhibiting a moderate effect size, with 80% power and using $\alpha = 0.05$ (*Tabachnick, 2013*). With an anticipated 40% response rate based on previous studies (*Alkhatib et al., 2015*), 240 pharmacies were invited to participate in the study, with the numbers of metropolitan and regional pharmacies in line with the

proportions of these in the sample frame (metropolitan community pharmacies made up 72.5% and regional community pharmacies 27.5%).

## Data analysis

Descriptive statistics (frequencies and percentages) were used to summarize the demographic profile of participants, and their responses. Responses on a five point Likert scale (i.e., strongly agree, agree, neutral, disagree and strongly disagree) were collapsed to a three point Likert scale (agree, neutral and disagree), for the purpose of analysis.

### Analysis of level of support for expanded prescribing

A composite score representing the respondents' overall attitude towards down-scheduling was obtained as the mean of the first seven statements assessing different aspects of the respondents' support for the down scheduling. The remaining two statements (relating to the design of the pharmacy, and whether OTC oral antibiotics would increase resistance to antibiotics) were not considered to be either supportive or unsupportive of down-scheduling, and therefore not included in the calculation of the overall attitude score. Being a mean of responses on a 1–5 scale, the composite score could be interpreted on a similar scale. Therefore, the overall response was classified as "in agreement" if the composite score was between 1 and 2 (inclusive), and classified as "not in agreement" otherwise. If the composite score was "in agreement", then the respondent was classified as "supporting rescheduling", and not in support of rescheduling if the score was "not in agreement". This variable was the primary outcome for the analyses. A logistic regression model was used to identify any demographic variables showing an association with this dichotomous variable. A backwards elimination strategy was used to find the 'optimal' model. In this approach, all the independent variables were initially included in the model, then the least significant was dropped, one at a time, until all variables remaining in the model were significantly associated with the outcome.

### Analysis of vignettes

An analysis of factors associated with the choice to refer the patients depicted in the case vignettes to their GP was analysed using a Generalised Estimating Equation (GEE). This model takes into account the correlation between vignette responses made by the same respondent (using an exchangeable correlation matrix structure). Inclusion of the vignette number as a factor in the model allowed a comparison of the referral rates between vignettes. The same model was used to identify factors associated with the appropriate oral antibiotic therapy selected for each vignette, except that cases where the respondent elected to refer straight to the GP were excluded from this analysis. Similarly to the logistic regression model above, a backward elimination strategy was used when fitting the GEE model. The final results are expressed as adjusted odds ratios, their 95% confidence intervals, and $p$-values.

A $p$-value <0.05 indicated a statistically significant association. The statistical analyses were performed using the SAS© version 9.2 software.

This study was approved by the Human Research Ethics Committee of Curtin University (Approval Number RDHS-04-15).

## RESULTS

Of the 240 questionnaires distributed (66 regional and 174 metropolitan), 90 were returned. Of these, 67/90 (74.4%) were from metropolitan and 23/90 (25.6%) regional community pharmacists. The response rates from metropolitan community pharmacists (67/174; 38.5%) and rural community pharmacists (23/66; 34.8%) were similar, leading to an overall response rate of 37.5% (90/240). Medium sized (based on turn-over of AUD\$1 m to \$2 m) community pharmacies were the largest cohort (38/89; 42.7%) (Table 1). Most pharmacists were in the age categories 31–40 years (32/89; 36.0%) and 21–30 years (31/89; 34.8%). It is evident (Table 1) that the overall number of respondents supporting the down-scheduling of selected oral antibiotics (as defined by the "composite") was 55.6% (50/90) which was independent of a wide range of demographic pharmacy and pharmacist characteristics.

### Respondents' level of support for an expanded prescribing role for oral antibiotics

With respect to pharmacist's views on "down-scheduling" of selected antibiotics, respondents strongly supported statements regarding: the use of their skills and knowledge (70/90; 77.8%), recognition by pharmacy clients (72/89; 80.9%) and treating of patients in a timely manner (72/90; 80.0%), as illustrated in Fig. 1.

Results from fitting the logistic regression model showed that no variables were significantly associated with the attitude towards rescheduling. The results indicate that approximately half of the respondents favoured down-scheduling regardless of any demographic variables such as their gender, age, or experience. This supports the univariate associations shown in Table 1. A total of 50/89 pharmacists (i.e., 56.2 %) estimated that on average 10 or more patients per week would seek advice for conditions where the pharmacist's best option would be to prescribe oral antibiotics, suggesting that they face these situations on daily basis.

The two statements which were excluded from calculation of the 'level of support for down-scheduling' score are included in Fig. 1. These show that the current layout of 69/90 (76.7%) of pharmacies would be conducive to diagnosis of infections and prescribing of antibiotics. Only 9/90 (10%) of respondents disagreed with the statement that provision of OTC oral antibiotics could increase resistance to antibiotics, with the majority (65/90; 72.2%) agreeing with this statement and the remaining 16/90 (17.8%) giving a neutral response.

### Respondents' level of support for community pharmacist prescribing selected oral antibiotics

More than 60% of respondents' (Fig. 2) supported expanded prescribing of phenoxymethylpenicillin (56/90), amoxicillin with clavulanic acid (55/88), flucloxacillin (61/88), cefalexin (64/90) and amoxicillin (66/90) with 70/90 (77.9 %) supporting trimethoprim. Few respondents supported rifampicin (2/87; 2.3%).

**Table 1  Demographic profile of the survey respondents (_n_ = 90), and their association with support for rescheduling of oral antibiotics.** _P_-values were obtained from the Chi-square test unless otherwise specified.

| Variable _n_ (%) | Number (%) | Number (%) supporting rescheduling | _p_-value |
|---|---|---|---|
| **Gender** (_n_ = 90) | | | 0.8504 |
| Female | 44 (48.9) | 24 (54.6) | |
| Male | 46 (51.1) | 26 (56.5) | |
| **Age group** (_n_ = 89) | | | 0.9166 |
| 21–30 | 31 (34.8) | 17 (54.8) | |
| 31–40 | 32 (36.0) | 19 (59.4) | |
| 41–50 | 10 (11.2) | 5 (50.0) | |
| 51 or more | 16 (18.0) | 8 (50.0) | |
| **Years registered as pharmacist** (_n_ = 88) | | | 0.5045 |
| 1–5 | 32 (36.4) | 17 (53.1) | |
| 6–20 | 35 (39.8) | 22 (62.9) | |
| 21 or more | 21 (23.9) | 10 (47.6) | |
| **Years worked in a pharmacy** (_n_ = 90) | | | 0.9149 |
| 1–5 | 25 (27.8) | 13 (52.0) | |
| 6–20 | 44 (48.9) | 25 (56.8) | |
| 21 or more | 21 (23.3) | 12 (57.1) | |
| **Position held in pharmacy**[a] (_n_ = 93) | | | |
| Proprietor | 31 (34.4) | 18 (58.1) | 0.7284 |
| Manager | 28 (31.1) | 16 (57.1) | 0.8386 |
| Pharmacist in charge | 23 (25.6) | 13 (56.5) | 0.9139 |
| Employed pharmacist | 6 (6.7) | 3 (50.0) | 1.0[c] |
| Consultant pharmacist | 3 (3.3) | 2 (66.7) | 1.0[c] |
| Other position | 2 (2.2) | 0 | 0.1948 |
| **Pharmacy setting**[d] (_n_ = 90) | | | |
| Group of shops | 24 (26.7) | 14 (58.3) | 0.7491 |
| City | 2 (2.2) | 2 (100) | 0.5006[c] |
| Neighbourhood | 23 (25.6) | 11 (47.8) | 0.3872 |
| Stand-alone | 4 (4.4) | 2 (50.0) | 1.0[c] |
| Next to doctor's surgery | 14 (15.6) | 9 (64.3) | 0.4744 |
| Regional shopping centre | 12 (13.3) | 7 (58.3) | 0.8352 |
| Medical centre | 10 (11.1) | 4 (40.0) | 0.3299 |
| Other setting | 1 (1.1) | 1 (100) | 1.0[c] |
| **Counselling room available** | 80 (88.9) | 45 (56.3) | 0.7461[c] |
| **Forward dispensing area** | 38 (42.2) | 24 (63.2) | 0.2147 |
| **Operation size (turn-over)**[b] (_n_ = 89) | | | 0.5137 |
| Small (<$1M) | 26 (29.2) | 13 (50.0) | |
| Medium ($1M−$2M) | 38 (42.7) | 24 (63.2) | |
| Large ($2M+) | 25 (28.1) | 13 (52.0) | |

**Table 1** (*continued*)

| Variable *n* (%) | Number (%) | Number (%) supporting rescheduling | *p*-value |
|---|---|---|---|
| **Pharmacy location** (*n* = 90) | | | 0.9139 |
| Metropolitan | 67 (74.4) | 37 (55.2) | |
| Rural | 23 (25.6) | 13 (56.5) | |

**Notes.**

[a]There were two respondents who classified themselves as consultant and proprietor pharmacists, and one as consultant and pharmacist in charge.

[b]One missing response.

[c]Fisher's exact test.

[d]Respondents may select one or more settings for their pharmacy.

## Case vignette scenarios

A total of 630 vignette responses were received from the 90 respondents (seven vignettes per questionnaire). Overall, responses to 426/630 (67.6%) of the vignettes were to treat the patient at presentation compared to the remaining 204/630 (32.4%) where referral to the GP was the preferred option. Of those who opted to treat, 380/426 (89.2%) chose to prescribe an oral antibiotic, with 334/380 (87.9%) of them selecting an appropriate antibiotic regimen. The decision to immediately treat was 80/90 (88.9%) for tonsillitis and 77/90 (85.6%) for otitis media. The level of immediate treatment was much lower for acute pyelonephritis 44/90 (48.9%). The GEE model to identify factors associated with the decision to refer the patient to their GP was based on all 630 responses to the case vignettes. Due to the large number of variables in the model, only pairwise interaction terms which would be expected to be correlated with each other were assessed for statistical significance (none was eventually included). Table 2 shows the full results of the GEE analysis. These data show that referral rates were similar for acute pyelonephritis, chlamydial urethritis and urinary tract infections (UTI) in pregnancy.

The difference in choosing to refer between pharmacists of different gender was small (male 33.5% vs female 31.2%) but statistically significant ($p = 0.0198$), with male pharmacists more likely to refer compared to female pharmacists. In addition, the older respondents were less likely to refer patients ($p = 0.0165$). Respondents from small turnover pharmacies ($p = 0.0122$) were more likely to refer than those from large turnover pharmacies; those who expected only a low number of patients to be seeking advice were more likely to refer, pharmacists with fewer patients treated at their pharmacy where oral antibiotics would be beneficial (<4 per week, $p = 0.0205$) and those respondents who were generally not in favour of expanding pharmacists prescribing role in antibiotics were also more likely to refer (Table 2).

### (a) Appropriateness of therapy selected by vignette respondents

A second GEE model was used to identify factors associated with an appropriate choice of antibiotic. This analysis used only the records where the respondent chose to immediately prescribe oral antibiotics (426 records). The final model (following the backwards elimination procedure) is shown in Table 3.

The reference category for comparing the different vignettes was the case of acute pyelonephritis, which had the highest rate of appropriate antibiotic use (95.5%). Compared with this group, antibiotics for otitis media ($p = 0.0060$), UTI in pregnancy ($p < 0.0001$)

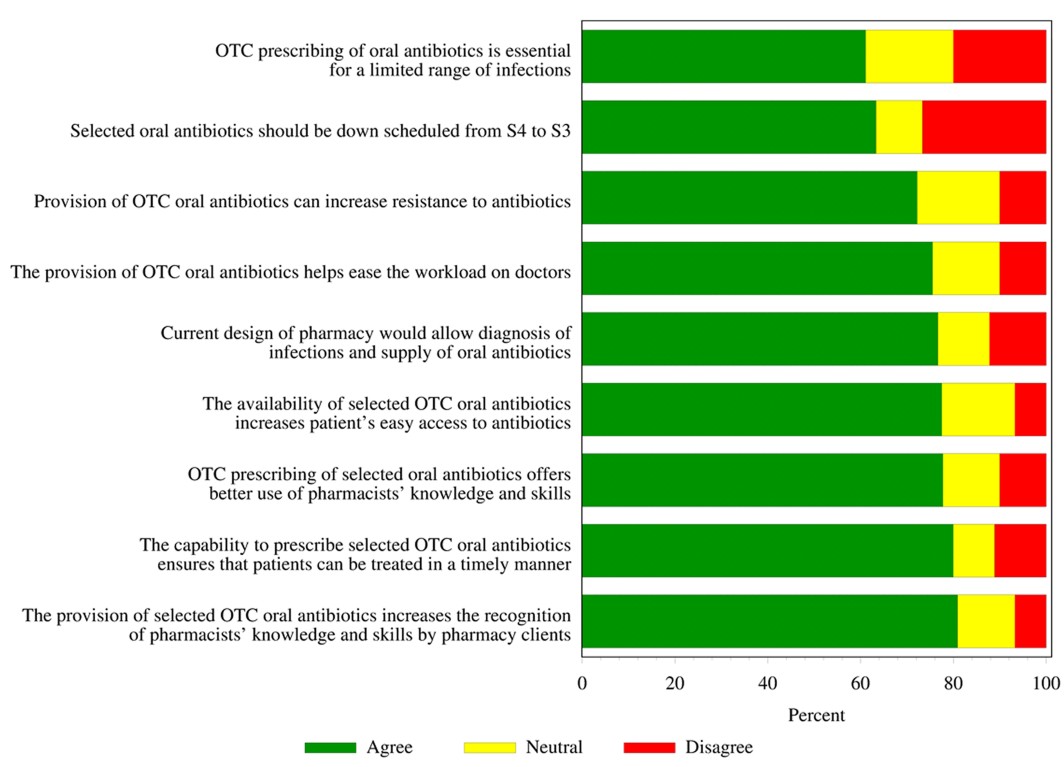

**Figure 1** Respondents' level of support for statements of views on down scheduling of oral antibiotics (*n* = 90) (OTC, over-the-counter).

and CAP (*p* = 0.0283) were significantly less appropriately prescribed. The appropriateness of prescribing for tonsillitis (*p* = 0.5146), chlamydial urethritis (*p* = 0.1337) and mild early cellulitis (*p* = 0.8528) were similarly appropriate to that for acute pyelonephritis (Fig. 3).

Older respondents (>50 years) were less likely to prescribe appropriately (70.8% vs 78.9%; *p* = 0.02). Similarly, respondents who were working in the role of a consultant pharmacist (only three respondents, responding to 13 vignettes), were less likely to prescribe appropriately compared to respondents holding other positions in the pharmacy (69.2% vs 77.5%; *p* = 0.0068).

### (b) Respondents' decision making following no improvement on initial therapy

Following three days of no improvement on initial therapy or 24 h for CAP (excluding immediate referrals to a GP), most respondents would refer the patient to a GP for all vignettes (Fig. 4) compared to selecting a different antibiotic, increasing the dose of the current antibiotic, using some alternative treatment or 'other'. For those who did not elect to refer to GP at this later stage, the most common choice was to select another antibiotic, in particular for otitis media (*n* = 12/14; 85.7%), CAP (*n* = 6/11; 54.5%) and UTI in pregnancy (*n* = 3/3; 100%) compared to other vignettes. In the case of CAP, respondents

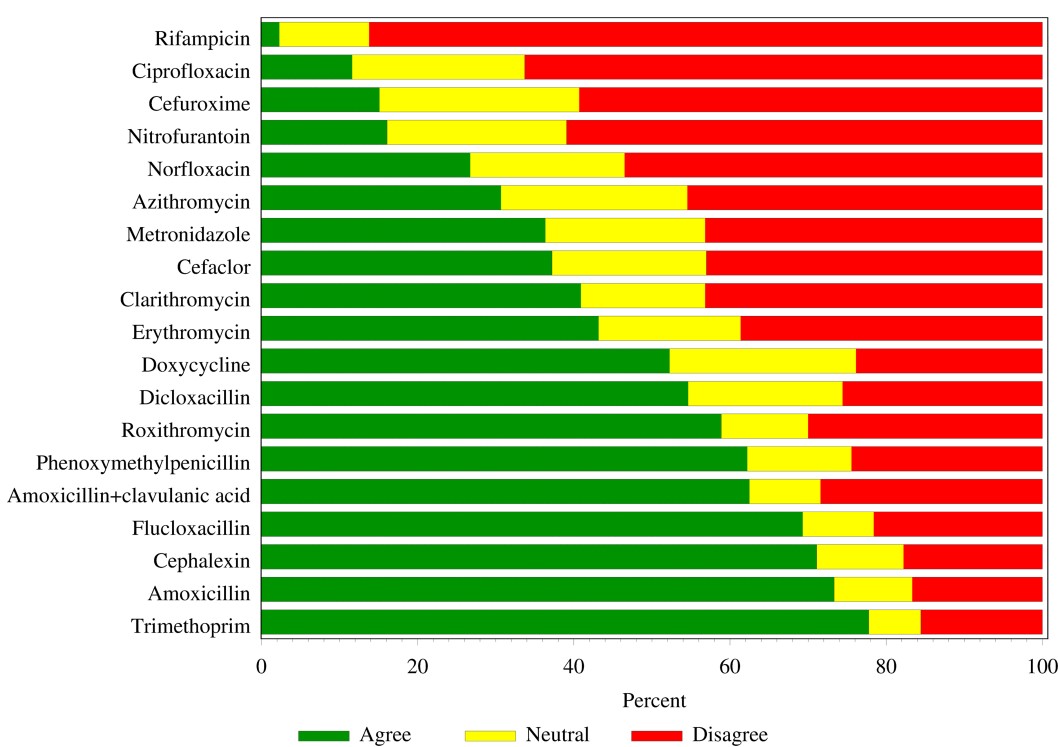

**Figure 2** Respondents' level of support for 'down scheduling' of specific oral antibiotics ($n = 90$).

were asked to select therapy after 24 h of no improvement instead of 3 days (*Antibiotic Expert Groups, 2014*).

## DISCUSSION

In the hypothetical situation that community pharmacists were permitted to prescribe OTC a range of antibiotics, this study reports their intended antibiotic prescribing behaviour when faced with scenarios simulating real practice. This study therefore provides a detailed insight into the appropriateness of their choice of prescribing for a graded range of infections, and identifies factors associated with appropriate prescribing. In addition it identifies medical conditions where the pharmacist would generally choose to refer the patient to their GP on initial presentation in the pharmacy, as well as following no symptom improvement. Acute pyelonephritis, chlamydial urethritis and UTI in pregnancy were significantly less likely to be treated by a community pharmacist than the other conditions presented.

In addition to findings suggesting a high level of appropriateness of antibiotic prescribing by pharmacists (334/380; 87.9%), this study also confirmed the existing literature indicating that pharmacists are supportive of an expanded prescribing role for a limited range of infections and antibiotics (*Hoti et al., 2010*; *Hoti, Hughes & Sunderland, 2014*). In this regard, pharmacists suggested a stronger preference for prescribing trimethoprim, amoxicillin and cefalexin and little support was suggested for prescribing antibiotics such as

**Table 2 Respondents' characteristics associated with their decision to refer a patient to their general practitioner (GP) initially (*n* = 630; results from the Generalised Estimating Equation model).**

| Variable | Number (%) referring to general practitioner | Adjusted odds ratio | 95% CI | *p*-value |
|---|---|---|---|---|
| **Case type** | | | | |
| Otitis media | 13/90 (14.4) | 0.13 | 0.06–0.27 | <.0001 |
| UTI in pregnancy | 37/90 (41.1) | 0.64 | 0.35–1.14 | 0.1279 |
| CAP | 32/90 (35.6) | 0.49 | 0.27–0.88 | 0.0168 |
| Tonsillitis | 10/90 (11.1) | 0.1 | 0.05–0.20 | <.0001 |
| Chlamydial urethritis | 45/90 (50.0) | 0.95 | 0.57–1.58 | 0.8479 |
| Mid early cellulitis | 21/90 (23.3) | 0.25 | 0.13–0.47 | <.0001 |
| Acute pyelonephritis | 46/90 (51.1) | 1 (reference) | | |
| **Gender** | | | | |
| Female | 96/308 (31.2) | 0.53 | 0.32–0.91 | 0.0198 |
| Male | 108/322 (33.5) | 1 (reference) | | |
| **Age group** | | | | |
| 51 or more | 23/112 (20.5) | 0.38 | 0.18–0.84 | 0.0165 |
| Up to 50 | 181/518 (34.9) | 1 (reference) | | |
| **Setting of pharmacy** | | | | |
| Medical centre | 34/70 (48.6) | 2.29 | 1.19–4.43 | 0.0137 |
| Other | 170/560 (30.4) | 1 (reference) | | |
| **Size of pharmacy (turnover)** | | | | |
| Small (<$1M) | 71/182 (39.0) | 2.38 | 1.21–4.69 | 0.0122 |
| Medium ($1M−$2M) | 88/266 (33.1) | 1.83 | 0.92–3.63 | 0.0837 |
| Large (>$2M) | 41/175 (23.4) | 1 (reference) | | |
| **Patients[a]** | | | | |
| Up to 3 per week | 33/63 (52.4) | 2.86 | 1.18–6.95 | 0.0205 |
| 4 or more per week | 171/567 (30.2) | 1 (reference) | | |
| **Rescheduling** | | | | |
| Neutral/Disagree | 113/280 (40.4) | 1.97 | 1.16–3.33 | 0.0116 |
| Agree | 91/350 (26.0) | 1 (reference) | | |

**Notes.**
[a]The estimated number of patients per week at pharmacy who would better be treated with oral antibiotics. Numbers are the number *n* of respondents and the percentage in parentheses.

rifampicin and ciprofloxacin. Pharmacists also indicated higher confidence in the treatment of less complicated infections. This is consistent with findings from previous Australian studies (*Hoti et al., 2010*; *Kay, Bajorek & Brien, 2006*; *Hoti, Hughes & Sunderland, 2014*). Notably, trimethoprim has been prescribed OTC in Scotland using a strict protocol providing improved patient access (*Booth et al., 2013*).

In their responses to the case scenarios provided, most pharmacists would refer patients following three days of no improvement on initial therapy (or following 24 h in the CAP case). This further supports the notion that community pharmacists are able to make decisions to treat minor uncomplicated infections without treatment delays and appropriately refer to the GP where complications arise. Minor ailments are reported to account for 10–20% of a doctors' workload (*Banks, 2010*) with doctors showing support

**Table 3** Variables associated with appropriateness of therapy selected (*n = 426*; results from the Generalised Estimating Equation model).

| Variable | Number correct (%) | Odds ratio | 95% CI | *p*-value |
|---|---|---|---|---|
| **Case type** | | | | |
| Otitis media | 59/77 (76.6) | 0.17 | 0.05–0.60 | 0.006 |
| UTI in pregnancy | 8/53 (15.1) | 0.01 | 0.00–0.04 | <.0001 |
| CAP | 46/58 (79.3) | 0.21 | 0.05–0.85 | 0.0283 |
| Tonsillitis | 74/80 (92.5) | 0.62 | 0.15–2.58 | 0.5146 |
| Chlamydial urethritis | 39/45 (86.7) | 0.37 | 0.10–1.36 | 0.1337 |
| Mid early cellulitis | 66/69 (95.7) | 1.16 | 0.23–5.83 | 0.8528 |
| Acute pyelonephritis | 42/44 (95.5) | 1 (reference) | | |
| **Age group** | | | | |
| 51 or more | 63/89 (70.8) | 0.44 | 0.22–0.88 | 0.02 |
| Up to 50 | 266/337 (78.9) | 1 (reference) | | |
| **Work position** | | | | |
| Consultant pharmacist | 9/13 (69.2) | 0.32 | 0.14–0.73 | 0.0068 |
| Other | 320/413 (77.5) | 1 (reference) | | |

for the diversion of management of minor ailments to non-medical prescribers including pharmacists (*Mansell et al., 2015*; *Bayliss & Rutter, 2004*).

In a majority (426/630; 67.6%) of the vignette responses in this study, the pharmacist opted to treat patients compared to referring them to the GP from the outset. As demonstrated in international studies (*Tonna et al., 2010*; *Law et al., 2012*), this level of responses to treat selected patient scenarios may suggest that community pharmacists are willing to manage a range of self-limiting infections. In addition, a large number of pharmacists followed the recommended guidelines leading to appropriate antibiotic therapy and potentially reduced antibiotic resistance (*Huttner et al., 2013*). An advantage of antibiotic management is immediate patient access. However, it is noteworthy that when mupirocin was reclassified OTC in NZ an increased resistance occurred in part from it being made available from pharmacies (*Upton, Lang & Heffernan, 2003*). However, similar levels of resistance were recorded in WA where it was only available on prescription (*Upton, Lang & Heffernan, 2003*).

It was found that pharmacists in this study who estimated that less than three patients per week would seek their treatment advice, were more likely to refer patients to the GP. A study in Spain reported that the number of hours worked and high patient numbers were associated with generating more prescriptions by community pharmacists, concluding that longer work hours was related to the pharmacist's understanding of their work (*Caamaño et al., 2004*). This is consistent with the present study where pharmacists from small turnover pharmacies were more likely to refer patients to the GP compared to large turnover pharmacies. This may indicate a more conservative approach to patient management than perceived by pharmacists in larger pharmacies. A study of community pharmacists in WA revealed that pharmacies with a large turnover were more supportive of the reclassification of chloramphenicol ophthalmic products than small and medium sized

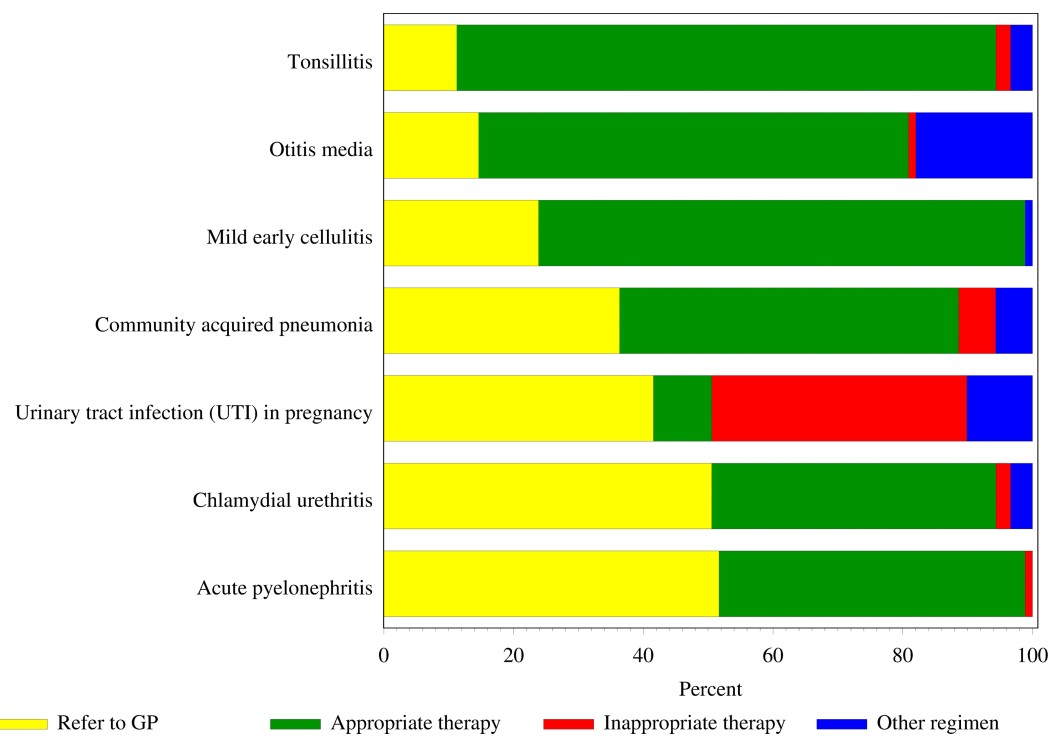

**Figure 3** Summary of respondents' choice to refer a patient to a general practitioner (GP) initially rather than treat with an oral antibiotic and the appropriateness of antibiotic selected (*n* = 426).

pharmacies (*Alkhatib et al., 2015*). An explanation for this may be stronger commercial interests for large pharmacies. Protocols would be essential to ensure antibiotic stewardship if selected antibiotics were reclassified.

Older respondents (>51 years) were less likely to refer patients to the GP and of those who chose to treat the patient directly, these older respondents were less likely to prescribe appropriately. Caamaño at al. suggested that the more experienced pharmacists generated fewer prescriptions and were more likely to refer a patient to their GP (*Caamaño et al., 2004*). These findings are supported by other studies where younger pharmacists were reported to place more importance on the patient's and pharmacist's autonomy compared to older pharmacists, describing older pharmacists as being 'more traditional' in their roles as pharmacists (*Pendergast et al., 1995*; *Isorna et al., 2004*). Another explanation may be the emphasis of education for older pharmacists would have been much less patient centred (*McWhinney, 1975*).

With a response rate of 37.5%, it is likely that some non-responders may have views on pharmacists' expanded prescribing of selected oral antibiotics which may differ from those based on the survey responders. However, with the high and consistent support for statements of views on prescribing oral antibiotics, high level of pharmacists' willingness to treat the patients in preference to GP referral, and overall high level of appropriateness of therapy selected, it is less likely that non-respondents' responses would have significantly influenced the findings. Furthermore, the wide range of conditions portrayed in the

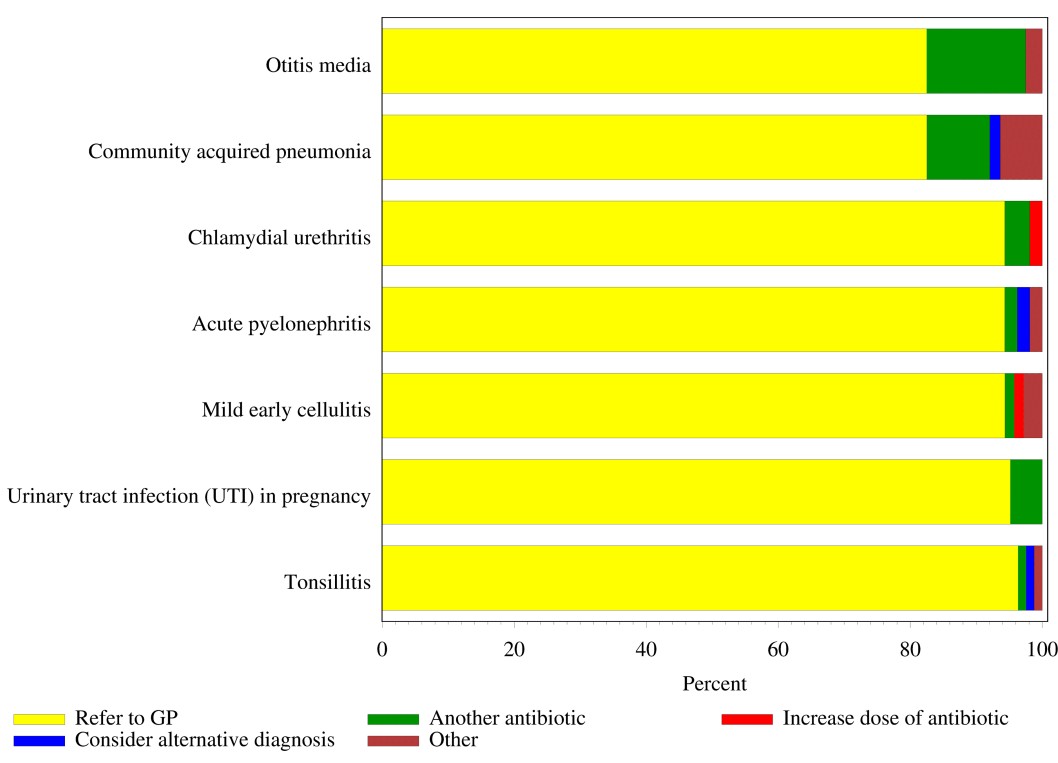

**Figure 4  Respondents' level of support for therapy following 3 days (or 24 h for community acquired pneumonia) of no improvement on initial therapy ($n = 426$).** Note: * 'Other' includes both drug and non-drug therapies.

seven vignettes strengthens the validity of the data. Case vignettes are often used to assess clinician's decision making behaviour and judgements (*Evans et al., 2015*). It should be highlighted that the literature suggests that this method provides predictive behaviour in circumstances appropriated by the vignette (*Evans et al., 2015*), thus the vignettes provide a demonstration of potential performance. A limitation of the vignettes is that, although face and content validity of the vignettes were determined, the grading scale was not specifically validated and therefore the reliability of the grading scale is not known.

This study demonstrated with the current framework of drug regulation that a "down-scheduling" option could be utilised to enable improved access for the public to a limited range of antibiotics for specific infections. This would also partially address the underutilisation of pharmacists' skills and their high accessibility. Appropriate methods would need to be developed for this to occur (*Reeves et al., 1999*).

Findings of this study should also be interpreted in context of the need to identify strategies and protocols that minimise antibiotic misuse in the community. Potential self-diagnosis and self-management of upper respiratory infections by the Australian community through use of antibiotic repeat prescriptions is undesirable (*Newby, Fryer & Henry, 2003*).

## CONCLUSION

In general, pharmacists indicated a high level of appropriateness of antibiotic selection when faced with a range of scenarios as vignettes. More complicated infections tended to be referred to the doctor. The findings of this study warrant consideration by professional bodies regarding expanding the role of pharmacists in the area of limited antibiotic prescribing for limited infections, as one of the strategies to addressing antibiotic misuse in the community and reducing unnecessary inconvenience and cost to the community. This would require the development of prescribing protocols that ensured appropriate prescribing.

## ACKNOWLEDGEMENTS

Authors wish to acknowledge all pharmacists who participated in this study,

### Funding
The authors received no funding for this work.

### Competing Interests
The authors declare there are no competing interests.

### Author Contributions
- Fatima Sinkala conceived and designed the experiments, performed the experiments, analyzed the data, contributed reagents/materials/analysis tools, prepared figures and/or tables, approved the final draft.
- Richard Parsons analyzed the data, contributed reagents/materials/analysis tools, prepared figures and/or tables, authored or reviewed drafts of the paper, approved the final draft.
- Bruce Sunderland and Petra Czarniak conceived and designed the experiments, analyzed the data, contributed reagents/materials/analysis tools, prepared figures and/or tables, authored or reviewed drafts of the paper, approved the final draft.
- Kreshnik Hoti conceived and designed the experiments, authored or reviewed drafts of the paper, approved the final draft.

### Human Ethics
The following information was supplied relating to ethical approvals (i.e., approving body and any reference numbers):

This study was approved by the Human Research Ethics Committee of Curtin University (Approval Number RDHS-04-15).

### Data Availability
Czarniak, Petra; Sunderland, Bruce; Parsons, Richard; Sinkala, Fatima (2017): Survey of attitudes of Western Australian pharmacists on rescheduling selected oral antibiotics. Curtin University. http://dx.doi.org/10.4225/06/5a01135fcf99e.

## Supplemental Information

Supplemental information for this article can be found online at http://dx.doi.org/10.7717/peerj.4726#supplemental-information.

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
