# Peer review of "A survey of the views and capabilities of community pharmacists in Western Australia regarding the rescheduling of selected oral antibiotics in a framework of pharmacist prescribing"

_PeerJ, doi:10.7717/peerj.4726_

## Round 0.1 · original submission · Major Revisions

While the reviewers were broadly positive about the manuscript, they all brought up a number of issues needing further detail or explanation. Please take the time to thoroughly address their questions and concerns before resubmission.

Reviewer 1 ·

Basic reporting

Thank you for the opportunity to review the manuscript titled “Exploring the views and capabilities of community pharmacists in Western Australia regarding the rescheduling of selected oral antibiotics in a framework of pharmacist prescribing”.

The researchers investigated the factors for appropriateness and the level of support for re-scheduling selected antibiotics to enable community pharmacist prescribing of oral antibiotics for a selected range of infections.

Discrepancies between the manuscript and the survey questionnaire raised a level of concern regarding the care taken over the conduct of the study and subsequent data analysis (additional comments below state where these discrepancies occur). Two of these discrepancies seem to be errors in the presentation of the survey question (Section B2) and case vignettes (Case 1 and Case 3) which may have affected participant responses. Light editing is required for clarity of some passages.


Abstract:
1. The denominator used (i.e. 630) refers to the number of responses (not respondents). Should be reported as percentage of responses (as each respondent provided 7 sets of responses for Section C).

Introduction:
2. Line 49-51: “There has been a movement towards the re-classification of selected prescription-only medicines to be made available from community pharmacies as S2 or S3.” Please cite reference/s for this statement.
3. Line 51-52: “Various level of control can be placed on selected Schedule 3 listings such as prescribing emergency contraception.” Please describe the levels of control alluded to in this statement for the benefit of international readers.
4. Line 55: “ … such decisions.” Please clarify.

Case Vignettes:
5. Page 4, line 101: “…if there was no improvement following 24 hours (otitis media) or 3 days (other vignettes).” The manuscript does not match the actual case vignette – it was Case 3 Community Acquired Pneumonia with acute respiratory symptoms that had the “After 24 hours of the treatment …” question; other vignettes stated “After 3 days of the treatment …”. See also item 13 of comments (under Results).

Questionnaire distribution:
6. Page 4, line 104 – please justify the 40% sample.
7. Page 4, line 109-110 – please explain how non-responders were identified. Were the questionnaires or reply paid envelopes printed with a unique code allocated to each community pharmacy?

Sample size distribution:
8. Page 4, line 115 – 240 pharmacies (each pharmacy may have one or more pharmacists).

Data analysis:
9. Page 5, line 125 – the manuscript did not match the survey – in Section B of the survey there were nine statements regarding “down-scheduling” of antibiotics. Please clarify.
10. Page 5, line 124-127 – please explain scoring used for the collapsed 3-point Likert in order to arrive at the classification of “in agreement” and “not in agreement”. How were neutral responses handled?

Results:
11. Page 7 and Figure 1 - Please explain why statements (h) and (i) (in Section B1 of the survey) were not reported in Results and in Figure 2.
12. Page 10, line 215-217 – Discrepancy: p-value for respondents in the role of consultant pharmacists stated as p=0.0009; whereas Table 3 shows 0.0068. Please clarify.
13. Page 10, line 226-227 – “In the case of otitis media, respondents were asked to select therapy after 24 hours of no improvement instead of 3 days (as per current guidelines).” If this is the case, then Case 1 and Case 3 vignettes in the survey were presented incorrectly to respondents (states 3 days for Case 1 and 24 hours for Case 3). This error may have affected how participants responded.
14. For the vignettes, were the responses for “Other (please specify regimen)” analysed? Were these responses appropriate?

Questionnaire:
15. Section B, Question 2 – responses to statements (from strongly agree to strongly disagree) are not aligned with the question asked (level of comfort in prescribing the listed antibiotics). There is a chance that respondents may have misread the question as the level of agreement that the listed antibiotics are re-scheduled, rather than their level of comfort in prescribing the listed antibiotics.

References:
16. Reference 15 and 17 are duplicates. Please remove one.

Tables and Figures:
17. Table 2: Formatting change – “Age group” to be in bold.
18. Figure 3 – not explicitly referred to in the manuscript.
19. Figure 4 – reconsider title (as one vignette had an interval of 24 hours instead of 3 days) e.g. Respondent follow up actions after an interval of no improvement with initial therapy.

Experimental design

The research question is well defined, relevant and meaningful. See also comments 5 to 15 provided in Basic Reporting section above.

Validity of the findings

Findings seem to be largely valid. Errors in the presentation of the survey question (Section B2) and case vignettes (Case 1 and Case 3) may have affected participant responses. Authors should clarify and propose how they could reasonably address this issue.

Additional comments

Congratulations on completing an interesting and useful study.

·

Basic reporting

Thank you for the opportunity to review this manuscript. The clinical decision making capability of pharmacists is an important area, particularly with the increasing scope of pharmacist practice. Should the editor choose to accept this manuscript for publication, the following comments should be addressed:

General: Overall this is a well written manuscript with few grammatical errors. Please clarify the author order as it differs from title page to abstract. Font changes occur, please review as it my do due to the pdf generation process. Please ensure all intext citations are consistently placed in reference to punctuation with reference to journal guidelines.

Title: Please add ‘survey’

Abstract: ‘Pharmacist’ not ‘pharmacy’ in results. Please add n values for all percentages. Provide the numbers compared with all p values.

Introduction:
Lines 33-6: Please reference statement.
Lines 38-41: Please reference statement.
Lines 49-51: Please reference statement.
Lines 51-52: Please clarify and reference ‘Various level of control can be placed
on selected Schedule 3 listings such as prescribing emergency contraception’.

References: Please review pagination consistency of citations. Reference 5 only reference with DOI. Reference 30 title is the only reference with an abbreviated title.

Experimental design

Methods:
Line 74: Mixed methods as defined by Creswell is the use of qualitative and quantitative methods to measure the same phenomenon. Looking at the attached questionnaire, the case vignettes have structured responses. As such, have two quantitative methods been employed in this instance? Further, are questions in Section B relating to attitudes and case vignettes in Section C which test knowledge the same phenomenon? These two issues would suggest multiple methods design rather than mixed methods. Please review.
Lines 79-82: who developed the questionnaire?
Line 98: were only ATG endorsed regimens provided? If not, how many options were endorsed of the total? Were all non-endorsed regimens classed as inappropriate? If so, who decided? How were ‘other regimens’ assessed?
Line 99: who graded the complexity? What criteria were used?
Line 104: why was 40% chosen as a sample? OK mentioned below. Please consider merging these sections so the power calculation is provided with the sample size.
Line 124: add possessive apostrophe to ‘respondents’.
Line 126-7: please clarify why “The overall response was classified as “in agreement” if the
mean score was less than (or equal to) 2, and classified as “not in agreement” otherwise.”
Line 129: change ‘best’ to optimal.
Line 134-5: Were all immediate referrals included in the GEE analysis? If so, please provide the research question being addressed as referral may be required or not required. Some vignettes could be self-limiting e.g. otitis media as opposed to pyelonephritis.

Validity of the findings

Results:
Line 152: Please add specify Australian dollars ‘AUD’.
Line 155: Why state approximately 50% support of down-scheduling antibiotics, yet Table 1 suggests more and the following statement in the abstract is made; “There was more than 60% respondent support for expanded prescribing (rescheduling) of commonly prescribed antibiotics.”
Line 157: Table 1: Please provide the question regarding support of downscheduling that is used for the cross tabulations in this table. Is it Selected oral antibiotics should be down scheduled from Schedule 4 to Schedule 3.? If so, why are the proportions in agreement for male (56.5%) and female (54.6%) both lower than the proportion reported in Figure 1? Why were multiple 2 x 2 Chi-squares and Fisher Exact tests used on nominal data with more than 1 degree of freedom i.e. position held, setting. Were Chi-square tests used for ordinal data?
Line 159-62: Please justify grouping these items with a collective average. I suspect there are multiple factors here. Was exploratory factor analysis performed? (see below).
Line 163: Figure 1: of interest is the large minority disagreeing with downscheduling that are not disagreeing with the potential benefits of downscheduling in following questions. This is a discrepancy worth mentioning. Why are only 7 of the 9 questions in this section of the questionnaire presented in this Figure? Please update.
Line 177-218: Please provide values that are being compared when a p value is reported.

Discussion:
Line 254-9: Please do not equate confidence with competence. While there is overlap, they are not the same.
Line 272-9: Please posit why conflicting findings were found regarding experienced pharmacists in different countries e.g. educational differences.
Line 286-90: Please consider the lack of observation or real-time deployment of case vignettes in this study. Would a case vignette questionnaire in which resources may be referred to and completion occurring at a time point of potentially low interruptions and work stress similar to real-life practice? Arguably these findings would be representative of optimal performance (sans training) rather than real-life performance.

Conclusion:
Well written.

Reviewer 3 ·

Basic reporting

The paper is easy to read with excellent English and an interesting topic.
The introduction would benefit from further work to provide context, ensure references support the statements made and provide balance. Line 28 and 29 is not supported by reference 3, and reference 2 is about secondary care – this needs to be clear in the introduction given it is a different context to this study which is about primary care, and also particularly given the perhaps excessively general statement attributed to it in line 32.
It is important to include references to community pharmacy research with respect to supply of antimicrobials by pharmacists, for example:
Booth JL, Mullen AB, Thomson DAM, et al. Antibiotic treatment of urinary tract infection by community pharmacists: a cross-sectional study. Br J Gen Pract. 2013;63:e244-e249.
Klepser ME, Adams AJ, Klepser DG. Antimicrobial stewardship in outpatient settings: leveraging innovative physician-pharmacist collaborations to reduce antibiotic resistance. Health security. 2015;13(3):166-173.
Gauld NJ, Zeng ISL, Ikram RB, et al. Antibiotic treatment of women with uncomplicated cystitis before and after allowing pharmacist-supply of trimethoprim. Int J Clin Pharm. 2017;39:165-172.
There may be others too, e.g. supplies of oseltamivir in the US I think, and New Zealand. While not being antibiotics they may be relevant.
Another relevant paper that I believe would be beneficial to include in the introduction is:
Reeves DS, Finch RG, Bax RP, et al. Self-medication of antibacterials without prescription (also called 'over-the-counter' use). A report of a Working Party of the British Society for Antimicrobial Chemotherapy. J Antimicrob Chemother. Aug 1999;44(2):163-177.
While old, this is a very well-considered paper that covers many of the key issues that still remain.
The paper would be more balanced and informative if there was a discussion of the downsides also of pharmacists providing these medicines, for example potential for overuse, both in the introduction and the discussion. Antibiotic resistance is a key concern internationally and in Australia, and there is a possibility of increased use of antibiotics under this scenario. Concerns about overuse of antibiotics are important with pharmacy availability. While not oral antibiotics, increased use is still relevant with chloramphenicol papers published internationally (2 in the UK and another in Australia I think), and the paper on mupirocin as below:
Upton A, Lang S, Heffernan H. Mupirocin and Staphylococcus aureus: A recent paradigm of emerging antibiotic resistance. J Antimicrob Chemother. 2003;51(3):613-617.
I would expect a mention (referenced) at least in the introduction of the rescheduling status of oral antibiotics without prescription e.g. the availability of azithromycin in the UK and trimethoprim in NZ, and possibly that trimethoprim and nitrofurantoin for UTIs in the UK did not progress.
The sentence 47-49 has references that are not particularly pertinent to the mention of S2 and S3 in Australia, are there more appropriate references? Papers by Gilbert et al, or Gauld et al, for example.
While the use of schedule 3 was explained and is well known in Australia, it would be helpful to use terms “pharmacy only” and “pharmacist only” throughout rather than S2 or S3, respectively, to make it easier for an international reader skimming the manuscript to understand quickly. See in particular line 51 for a mention os S2 and S3. My understanding, which might not be correct, was that there needed to be direct involvement of a pharmacist, with S3 medicines in Australia, if legislation or guidelines allow for supervision instead it would be very helpful to include that reference (line 36-38).
There is quite a difference between an specialist antibiotic pharmacist in hospital and a community pharmacist with no additional training. Would additional training be recommended, as is required for trimethoprim in NZ and is encouraged for azithromycin in the UK? Would there be a requirement or expectation that Australian guidelines would be adhered to?
The tables should include the abbreviations used in full at the bottom of the table.
Expanding line 54 on the potential benefits of such availability would provide context.
The English is excellent throughout and the paper is easy to understand. Check the wording of the sentence line 266-267.
Line 270-271 – I agree with the potential for commercial interests, perhaps larger pharmacies might also have more pharmacists on staff to make it easier to manage longer consultations. Noting this commercial interest though, is there a risk for inappropriate supplies, e.g. immediate treatment in someone with minor symptoms who could take a wait and see approach? Supplies in advance of need for “in case”. Some studies in Australia have shown S3 medicines being sold without pharmacist input – this should be mentioned (referenced) and consideration of how the risk might be mitigated would be useful.
It would be helpful in the discussion to think about how this availability could work in practice, for example, by simple down-scheduling, or by allowing pharmacists to become prescribers of antibiotics if they have completed special training, perhaps in a similar way as for vaccinations in Australia. Bear in mind the codeine example whereby upscheduling is about to occur in Australia after some misuse and abuse, despite the S3 category and resultant patient harm.
There would be benefit in considering further research which is required. In particular, research on actual behaviour in the UK with azithromycin could help to inform the appropriateness of therapy.

Experimental design

The research is original. The aim is well written and meaningful.
Method: The way the vignettes were developed, based on literature and by using the Australian Therapeutic Guidelines appeared to miss expert input specifically into the vignettes e.g. from an infectious diseases specialist or consultant microbiologist and a doctor working in general practice. I think this is an important deficiency. For example, it seems surprising to suggest treating pyelonephritis in pharmacy. This can be a serious illness and at best cystitis would be treated in the pharmacy and pyelonephritis referred to a doctor given that pharmacists are not trained to the same degree as doctors. Either this was done and was overlooked in the method, or it is a limitation that should be mentioned and affects the usefulness of the research in my view.
Another point is that the right antibiotic is important for antibiotic resistance, but so is the correct dosing and the correct duration (e.g. Dryden). It would be helpful to mention in the limitations that this was not measured and is relevant.
The questions asked about their opinions of down-scheduling avoided asking about any negative areas, for example, OTC prescribing of antibiotics could increase the risk of antibiotic resistance. Or OTC prescribing of antibiotics could lead to unrealistic expectations from patients of having antibiotics when they are not appropriate. Having a consultant microbiologist involved would have helped provide more balance. This could be an area to highlight for further research, and I think should be mentioned as a limitation - that the potential downsides were not explored.

Validity of the findings

See notes under experimental design about the development of the vignettes.
Data is fine.

---

## Round 0.2 · Minor Revisions

Please address the remaining minor concerns/edits presented by both reviewers. In particular, please address those comments regarding the case vignettes.

·

Basic reporting

No further comments

Experimental design

No further comments

Validity of the findings

Two comments remain, specific to the case vignettes.

1. The deployment of case vignettes via non-supervised survey is a limitation to external validity of actual practice as previously commented. Despite responding that my initial comment related to a referenced statement, please note what the reference 27 (Evans et al, 2015) states "Similarly, in regard to the external validity and generalizability of findings, participants’ behavior in a vignette study is not intended to be interpreted as representative of their behavior in the real world, but rather as strong predictors or proxies for such behavior, given the circumstances approximated by the vignette." Therefore my assertion that the vignettes are representative of optimal performance (sans training) rather than real-life performance stands as the vignettes may be answered in a situation in which resources may be referred to and completion may occur at a time point of low interruptions and work stress. The authors can simply state that the vignettes provide a demonstration of potential performance. My apologies for not making my meaning sufficiently clear in my initial comment.

2. As previously commented and as Reviewer 3 eloquently commented, the content validity and grading criteria of the vignettes themselves may be questioned. Piloting with community pharmacists determines face validity. The content validity is questionable as the process of vignette development by the researchers themselves and piloting with community pharmacists with extensive experience with antibiotics may not be deemed equivalent to content experts such as a panel of specialist infectious diseases physicians. A seperate panel of experts in addition to piloting is a recommendation of Evans et al, 2015 (reference 27). Further, the method of determining the grading has not been provided or validated. The reliability of the grading scale is not known. Again, these issues could simply be expressed as limitations. These limitations do not significantly impair the quality of the overall work but nonetheless should be stated.

Additional comments

Thank you for the extensive revisions. The revised article is an excellent and complex piece of research and should be a significant contribution to the area.

Reviewer 3 ·

Basic reporting

The abstract contains abbreviations which are not in full.
Line 142 were not was
Line 274 remove the apostrophe

Experimental design

No comment

Validity of the findings

No comment. Pleased to see the addition of protocols.

---

## Round 0.3 · accepted · Accept

Thank you for addressing the last set of reviewer concerns and congratulations again.